# Building Connections and Striving to Build Better Futures: A Qualitative Interview Study of Alcohol Recovery Navigators’ Practice in the North East of England, UK

**DOI:** 10.3390/ijerph22010111

**Published:** 2025-01-15

**Authors:** Domna Salonen, Amy O’Donnell, Katherine Jackson, Sarah Hulse, James Crosbie, Ryan Swiers, Fiona Tasker, Gemma Muldowney, Anna Pickford, Floor Christie-de Jong, Eileen Kaner, Emma-Joy Holland

**Affiliations:** 1Population Health Sciences Institute, Newcastle University, Newcastle upon Tyne NE1 7RU, UK; d.salonen2@newcastle.ac.uk (D.S.); amy.odonnell@newcastle.ac.uk (A.O.); kat.jackson@newcastle.ac.uk (K.J.); eileen.kaner@newcastle.ac.uk (E.K.); 2North East North Cumbria Integrated Care Board, Sunderland SR5 3XB, UK; sarah.hulse1@nhs.net (S.H.); jamescrosbie@nhs.net (J.C.); ryan.swiers@nhs.net (R.S.); a.pickford@nhs.net (A.P.); 3North of England Commissioning Support (NECS), Durham DH1 3YG, UK; 4South Tyneside and Sunderland NHS Foundation Trust, Sunderland SR4 7TP, UK; 5ReCoCo-Newcastle Recovery College, Newcastle upon Tyne NE1 6UF, UK; fiona@recoverycoco.com (F.T.); gemma@recoverycoco.com (G.M.); 6School of Medicine, Faculty of Health Sciences and Wellbeing, University of Sunderland, Sunderland SR1 3SD, UK; floor.christie@sunderland.ac.uk

**Keywords:** alcohol, mental health, health inequalities, care navigation, continuity of care, integrated care, alcohol care teams, qualitative research, self-determination theory

## Abstract

To address the holistic and continuity of care needs of people who attend North East hospitals frequently for alcohol-related reasons, Recovery Navigator (Navigator) roles were introduced into Alcohol Care Teams in six hospitals in the North East of England, UK, in 2022. The Navigators aimed to provide dedicated holistic support to patients experiencing alcohol harms, starting whilst in the hospital with the potential to continue this beyond discharge. This qualitative study explores the contributions that the Navigators make towards integrated alcohol care. Twenty-five semi-structured interviews were undertaken with 7 patients, 1 carer, and 17 staff. We used reflexive thematic analysis and applied the concept of continuity of care and Self-Determination Theory. The findings suggest that all of the participants value Navigators having dedicated time to work with patients to address their social needs, that patients benefit from having someone who provides relational support and is ‘gently persistent’, and that most of the Navigators have good relationships with community providers and have supported the transition of patients to these services. Staff recognise the challenges of holistic alcohol care in hospitals, and the support of the Alcohol Care Teams and Navigators is seen as invaluable. Navigators help to address gaps in the provision of holistic support for patients who experience significant health inequalities.

## 1. Introduction

Alcohol use is a leading risk factor for premature mortality and chronic disease, accounting for over three million deaths per annum globally [1]. Alcohol is also closely linked with health inequalities, with people living in relative deprivation experiencing greater alcohol-related harm than people with high socioeconomic status, even when they consume the same amount or less alcohol [2,3]. In the United Kingdom (UK) specifically, there has been a steep rise in alcohol-related mortality since the COVID-19 pandemic, particularly in more deprived regions. The North East of England, in particular, faces challenges due to experiencing poorer health, economic and social outcomes, and higher than average levels of deprivation than other areas of England [4]. The North East has the highest alcohol-specific mortality rates in England [5], while alcohol-related deaths have increased by 22.9% since 2019 [6]. Alongside the individual and wider societal impacts, alcohol-related harm also places a heavy economic burden on the UK healthcare system, with the cost to the NHS estimated to be approaching GBP 5 billion a year for England alone [7]. In the North East and North Cumbria regions of England, 39% of men and 18% of women are estimated to drink at high-risk levels [8].

People who attend hospitals for alcohol-related reasons comprise a significant proportion of those accessing emergency departments (EDs) in the UK; more than in either general hospital wards or inpatient psychiatric units [9]. There is a particular group of individuals who attend EDs frequently for alcohol-related reasons who also have mental health disorders and long-term physical conditions, live in isolation or poverty, and have attempted or contemplated suicide [10]. Consequently, people who attend EDs frequently due to alcohol-related reasons can present with a range of psychosocial support needs, alongside immediate biomedical concerns [11]. In previous qualitative research, people who have simultaneous mental health symptoms and alcohol-related challenges [12], and people who attend hospital frequently due to alcohol-related reasons [11], have shared that relationship-building with practitioners, combined with condition-specific and holistic support, can facilitate good care experiences and treatment outcomes.

Clinical guidelines [13] recommend a comprehensive assessment of a person’s overall health profile and social situation to address alcohol-related risk and harm effectively. In England, the NHS Long Term Plan [14] led to the introduction of hospital-based Alcohol Care Teams (ACTs) in the areas experiencing the greatest alcohol harm. The presence of a specialist alcohol nurse and the multidisciplinary care provided by ACTs has proven benefits, with reduced alcohol-related mortality [15] and earlier intervention for alcohol-related health problems with clear, centrally managed alcohol care pathways [16].

Recovery from an alcohol use disorder (AUD) is a process towards reduced drinking or abstinence, and often involves changes in social support, psychosocial functioning, and wellbeing [17]. There is yet no refined understanding how recovery capital [18]—an individual’s capabilities and material and social resources—impacts recovery, although it is important to remember that different factors might not have equal weight in recovery outcomes [19]. Life narrative interviews with people in addiction recovery illuminate how gaining relationships, skills, and new purpose in life are important in maintaining recovery [20]. Therefore, while ACTs might be a key part of the solutions towards better alcohol care in hospitals in the areas that are most impacted by alcohol-related health inequalities, recovery support towards improved health and wellbeing needs to extend beyond medical environments. Current information governance arrangements do not always permit information to be shared between NHS hospitals and non-NHS community-based recovery support services [16]. This can lead to disruptions in care and support for people who attend hospitals frequently for alcohol-related reasons.

In the North East and North Cumbria (NENC) region in England, a new role of Recovery Navigators (“Navigators” from here onwards) was established in Alcohol Care Teams in six acute hospital settings from 2021 onwards. Navigators’ roles involve care co-ordination across the health and social care. They work to support continuity of care and provide holistic support to people who attend hospitals frequently for alcohol-related reasons. Although Navigators work within hospitals, they can have non-clinical backgrounds, and their job description does not involve traditional clinical support staff competencies.

While the Navigators are new in NENC ACTs, similar roles have been introduced internationally in a range of health and social care contexts, with an aim to improve continuity of care and health and wellbeing within specific populations with multifaceted support needs [21,22,23,24,25]. In rural British Columbia, the introduction of Navigators improved the access to comprehensive support for people with addictions and mental health problems [23]. In Glasgow and Edinburgh, Scotland, UK, Navigators built therapeutic rapport with people who attend EDs after episodes of violence and supported them to connect with community services that can help with their long-term health and wellbeing needs [24]. In different contexts, Navigator roles can have different requirements for continuous professional development due to them being delivered by registered health and social care professionals in some settings, and people without formal qualifications or prior experience in others [25]. Health Education England [26] points out that care navigation is, to some extent, a key activity for all health and social care staff, while some practitioners take bespoke roles as Navigators. Required levels of competence, as outlined in their competency framework, apply to each Navigator depending on the nature of their role. The Navigator roles in NENC ACTs correspond with the “enhanced” level—working with independence and having strong interpersonal communication skills, but not leading teams or managing complex operations.

Navigators and ACTs in the NENC are an example of wider developments in the mental healthcare workforce, including addictions and AUDs. Care navigator roles are conceptually similar with those of social prescribers, who, in the UK, go by various names, e.g., “link workers”, “navigators”, and “wellbeing co-ordinators” [27,28]. Peer workers [29,30,31,32] work similarly with care navigators, providing person-centred support and helping people with a range of health and social support needs to access relevant support. Peer workers will have personal experience of similar challenges to the people who access their service [33]. Before the development of ACTs, psychiatric liaison services were introduced in general hospitals and their EDs to address the needs of patients who visit hospitals for mental health reasons [34,35]. Navigators within NENC ACTs are entering this landscape of services that have conceptual and practical differences and overlaps, with an aim to improve post-discharge support for people who attend hospitals frequently for alcohol-related reasons.

This envisaged better support requires the consideration of care across services, where Navigators have a key role in facilitating contact between services and patients. Continuity of care (CoC) refers to the patient’s experience of consistent and coherent care over time and multiple individual encounters with healthcare providers [36]. CoC can be understood through the following three overlapping processes: relational, informational, and management continuity [36,37]. Relational continuity considers the direct relationship between a patient and a practitioner. These trusting relationships are the foundation for any other type of care continuity. Informational continuity is about communication between patients, family members, and practitioners across the health system to ensure everyone involved in care has up-to-date key information. Management continuity refers to how care is co-ordinated across practitioners, services, and organisations.

However, people’s motivations and choices need to be understood in a wider context beyond service encounters. Self-Determination Theory (SDT) is a developmental theory of human motivation and wellbeing [38] that recognises that health outcomes develop in a relational context. SDT views every person to have intrinsic motivation, i.e., it is in people’s nature to learn, explore, and make the most of their abilities. However, intrinsic motivation requires suitable conditions to flourish. According to SDT, to attain wellbeing, an individual’s basic needs must be satisfied; this includes autonomy, competence, and relatedness [39,40]. People can realise their potential when they have a choice over what they do (autonomy), when they can use and develop their skills (competence), and when their past and current relationships are a source of security (relatedness). Meeting basic psychological needs can lead to enhanced self-motivation and wellbeing, whereas unmet basic psychological needs can have an opposite diminishing effect [38].

In this paper, we explore the views and perspectives of patients, carers, and staff about the introduction of Navigators in North East England. In doing so, we aimed to understand how this new role contributes to integrated care for people who attend hospitals frequently for alcohol-related reasons and to develop insights into how Navigators contribute to therapeutic change in patients’ and carers’ lives. We have drawn on a conceptual and theoretical approach that helps to illuminate the specific aspects of care processes and patient experiences, namely, continuity of care (CoC, [36,37]) and Self-Determination Theory (SDT, [38]). Both CoC and SDT acknowledge the intrinsic role of relationships between patients and practitioners as a precondition for good care and health outcomes.

## 2. Methods

This qualitative study was conducted as part of a wider mixed-methods evaluation of the introduction of Navigators, which also employed retrospective and prospective analyses of routine healthcare data, and a survey of healthcare professionals’ attitudes towards alcohol as a health problem. Here, we present the qualitative study process and findings only.

The aim of the qualitative study was to understand the Navigators’ contributions to the care and recovery of people who attend hospitals frequently for alcohol-related reasons. This understanding was developed by exploring the experiences of Navigators and their colleagues, patients, and carers.

### 2.1. Setting

The research was conducted in five acute hospitals across the NENC, located in predominantly urban areas; the sixth hospital had a vacant Navigator post throughout the study. These hospitals provide care to patients across a wide geographical catchment area, ranging across urban, rural, and coastal neighbourhoods, and different levels of socioeconomical deprivation and affluence. The wider alcohol care infrastructure around and within these hospitals varied significantly depending on local-level commissioning arrangements and the capacity of the NHS and third-sector alcohol services and groups. For example, one hospital had an Alcoholics Anonymous group that met on hospital premises and was open to both inpatients and other group members. In another hospital, a charity was preparing to offer in-hospital services for patients with addictions to substances other than alcohol. In two other hospitals, staff were the main connection to surrounding community-based services through referrals and liaison by phone and e-mail. One site had data protection agreements that allowed specific community- and hospital-based staff to access records across both services.

### 2.2. Sample and Recruitment

We sought to recruit a maximum variation sample of individuals with experience of the work carried out by Navigators, including patients, carers, and staff. For patients and carers, we aimed to recruit both men and women from all of the involved NENC geographical areas across an age range of 18–51+. For staff, we aimed at recruiting Navigators and their multidisciplinary colleagues with frontline and senior management roles, across statutory and voluntary services and hospital and community settings, and within all of the involved NENC areas.

Navigators supported patient and carer recruitment in the participating NHS Trusts, providing potential participants with study information and sharing patient contact details, with consent, to the research team via an encrypted e-mail. Patients could also choose to contact the research team directly. In addition to the study participation leaflet, we developed a shorter one-page information sheet and a YouTube video to support recruitment through a range of engaging and informative materials. Potential staff participants were identified through ACT managers and Navigators. Researchers approached them by e-mail. Participant information sheets and consent forms were sent to staff participants who expressed an interest in participating. Researchers did not make any of the gatekeepers aware of which staff members had participated in the study.

### 2.3. Patient and Public Involvement and Ethical Approval

Three Patient and Public Involvement and Engagement (PPIE) groups were established to reflect the geographical areas of the hospitals. Two groups involved experts by experience recruited from recovery charities located in two North East cities; these groups met primarily face-to-face. The third PPIE group met online and involved both people with carer experience and a public member with an interest in health research. Two PPIE members joined the project management group that met monthly and included researchers, practitioners, and commissioners.

PPIE members contributed to the study throughout its delivery, including providing feedback to guide the development of engaging research recruitment material, contributing to analysis of data, and co-authoring a lay summary of the research findings for wider dissemination.

Ethical approval for this study was given by the NHS Health Research Authority (IRAS 315158).

### 2.4. Data Collection

Semi-structured interviews were used to explore the role of Navigators; the perceived impacts their practice had on patients’ health, wellbeing, and engagement with wider support services; barriers and facilitators to engagement between the Navigators and patients/carers; and thoughts about improving the service. Separate loose topic guides were developed for staff, patient, and carer interviews to help focus discussions with participants. Researchers probed to encourage participants to share their insights, to elaborate on emergent issues, and to clarify terms or meanings as necessary [41,42]. Two researchers (E.J.H. and D.S.) carried out all of the interviews.

Patient and carer interviews took place in-person, with informed consent, and participants completed a demographic information form to record age, gender, ethnic group, educational level, marital status, and employment status. Interviews lasted for 34 min on average. Patient and carer participants were offered a GBP 20 voucher to thank them for their participation. Given the potentially sensitive and/or distressing nature of the interview discussion, D.S. or E.J.H. contacted them by phone the following week to ensure participant wellbeing. Staff interviews took place online using video conferencing software (MS Teams). All of the interviews were audio-recorded and transcribed verbatim prior to anonymising and importing the transcriptions to NVivo 14 for coding.

### 2.5. Analysis

Reflexive thematic analysis was applied, including familiarisation with the data, coding, generating, developing, refining, and naming the themes, and writing up [43]. D.S. is an occupational therapist with experience of providing a person-centred service in psychiatric inpatient services and acute hospitals, including follow-up care in the places where people who have needed hospital care live their everyday lives. She has worked regularly with people with AUD in combination with other conditions. E.J.H. is a white British researcher with a psychology background. She has experience with interviewing adults with AUD and other conditions, but no direct experience of this herself. Both of the researchers aspired to maintain a neutral curiosity to enable learning about Navigator roles, with the aim to improve the health and wellbeing of people who access hospitals frequently for alcohol-related reasons.

The researchers met regularly to discuss reflections about the data, codes, and theme development, and kept shared reflexive notes. They initially both coded four transcripts independently, and then discussed the initial codes and coding approaches. These transcripts were further discussed and refined at a subsequent Patient and Public Involvement and Engagement (PPIE) workshop, in which the researchers facilitated small group activities where PPIE partners and researchers read, discussed, shifted, and sorted quotes from the transcripts. The researchers kept notes to capture insights from the conversations and to record potential codes. At the end of the workshop, the researchers shared their initial codes. PPIE partners identified connections between the different codes and shared further reflections about the data. After this, the researchers agreed on a broad coding frame. Researchers met regularly and maintained a shared reflexive document when revising the codes and developing the themes. Codes were continuously added, deleted, and merged. We observed that motivation development and social connectedness were key elements across all of the themes. This guided the next steps to refine the analysis and connect the work to the wider theoretical landscape through SDT and CoC.

## 3. Results

In total, 17 staff, 7 patients, and 1 carer took part in the study. The staff participants were from five NHS Trusts and their connected community services. These included Navigators, Alcohol Care Team leads, specialist nurses in safeguarding and mental health assessment, and practitioners and managers in community-based addiction services. Patient participants were recruited through three NHS Trusts. The two patient participants and the carer participant were female. The age range of the patient and carer group was 34–69 years, they were all white British, and had a mixture of educational backgrounds ranging from leaving school before completing the GCSEs/O-levels to having a higher degree. Involvement with the Navigator indicated a history of alcohol-related hospital admissions. For this reason, patterns of alcohol use were not asked about in the demographic form or interview questions directly. However, during the interviews, all of the patient participants shared, unprompted, that they had given up drinking any alcohol.

While all of the Navigators had received the same role specification through the Integrated Care System and developed the local job description based on this, in practice, there was some variation in the service delivery and the eligibility criteria for the service. For example, in one Trust, the Navigator worked with any patient who had a background in alcohol use, and, in another Trust, the Navigator only worked with patients with five or more hospital attendances in the past year.

We developed five themes across the participants’ accounts. The first four themes capture aspects of the Navigators’ practice which relate to supporting the recovery of patients, while the fifth theme relates to the support that the Navigators were perceived to need to work effectively and safely. All of the names used are pseudonyms.

### 3.1. Listening to Patients Who Used to Come and Go Unheard

Practitioners who worked in EDs suggested that, before the introduction of the Navigators, there was no staff member who had a dedicated role to work with patients with alcohol-related challenges. Moreover, the practitioners suggested that, despite the fact patients attending hospitals frequently for alcohol-related reasons were recognised as having a range of psychosocial needs, the ED was limited in its capacity to address these wider support issues.

The thing about A&E is it’s emergency medicine, and their thought process, as it should be, starts and stops with emergency medicine… I can understand why they didn’t look at the bigger picture [for patients] because that’s not necessarily their priority and it probably never will be.(Diane, Alcohol Care Team)

The following quote from a doctor working in an ED illustrates a shared perception amongst many of the participants that Recovery Navigators were now helping to address this care gap:

…the fact that there is a team that has time to dig a bit deeper and figure out actually what has caused you to come in is really important. Because as much as I’d love to have half an hour or an hour to speak to each patient that comes in for their social issues, you just don’t have the physical time.(Henry, ED doctor)

Participants’ accounts highlighted some of the other ways in which the Recovery Navigators were viewed as helping to provide a hospital environment that could better meet the needs of people who attend regularly due to alcohol-related reasons. For example, and drawing on SDT, by better addressing patients’ basic psychological needs, specifically, autonomy, relatedness, and competence, the Navigators appeared to be enabling more open and honest conversations with patients, as the quote below illustrates:

…because you feel like you can open up, and tell them everything, then they know the best path for recovery. Rather than sort of saying, “Oh well, I don’t drink that much…” you know, like, sugarcoating it, or not being completely truthful about how much, or anything like that. So they make it very easy to be very, very truthful, and, like I say, without any judgement, which is really good.(Alice, patient, 40–45)

As Alice’s account conveys, building an environment where patients can share their experiences more openly appeared to be helping better meet their needs of relatedness in care provision. Additionally, the participants’ accounts suggested that the relational way in which Recovery Navigators worked was also helping to promote increased patient autonomy over treatment outcomes, as Derek’s quote below highlights:

[the Navigator], like the full team, put the effort in with you but if you don’t respond and don’t want to engage, it’s not going to work.(Derek, patient, male, 56–60)

### 3.2. Working Where Navigators Are Needed Most

The original vision for the Navigator role was that they would provide an assertive outreach service and work across the boundaries of hospital and community services. However, we found that, in practice, the Navigators had adapted their model of care to suit the available local context and resources. In most of the sites (four out of five NHS Trusts), the hospitals were the only place that the Navigators and patients were meeting face-to-face. Only one of the Navigators was actively visiting patients in the community, and most of the patient contact post-discharge took place by text messages and phone calls.

Most of the practitioners who worked closely with the Navigators appeared to consider hospitals to be the most appropriate place for their work. They particularly valued how the Navigators were helping to reduce time pressures for other types of staff working in the region’s hospitals. Moreover, the Navigators’ consistent presence in the hospital was viewed as having therapeutic value for patients, as the quote below illustrates:

They’re in hospital. And they may have been pondering, thinking, reflecting in the middle of the night when they cannot sleep, that sort of stuff, “What can I do to get myself better? What can I do?”... you have people like [Navigator] and her team, who come up to the bed and go, “Hello. I’m from the alcohol team. Can I have a conversation please?” Do you know what I mean?(John, specialist nurse)

These views were echoed in patient participants’ accounts, who also appeared to see the hospital as a good environment for the Navigators to work. In the quote below, Barry shares his reflections of the crucial importance of the Navigators being there to meet people when they are in the hospital, whilst acknowledging the need to prioritise certain patients over others:

… the amount of times I’ve been in hospital now, and no matter how bad I’ve been, there will be somebody in who is in an infinitely worse state than I am. Obviously her priority is to be, you know, with the people on the ward.(Barry, patient, 50–55)

Another patient participant, Alice, reasoned that, as she was able to access high-quality community recovery support in her local area, the Navigator’s time was best used in the hospital to enable the initial contact with treatment services. Looking forward, it was conveyed that, in two NHS Trusts, the intention was to transition into a model where the Navigator worked across the hospital and the community, i.e., the patients’ everyday environments, such as their homes and local community settings. However, one Navigator expressed the following concern about the potential for duplication and confusion between the role played by Navigators and support workers in community-based services:

So, I don’t know if it would be worth having a sit down with management there and thinking about, “Okay, where do we meet in the middle?”… Because I think it would just be really confusing for the patient. They wouldn’t know which worker to liaise with for what if they’ve got two, and it just might be too much of the same. So, I don’t know, maybe external agencies need to be part of that meeting, and figuring out a middle ground, maybe.(Hannah, Recovery Navigator)

### 3.3. Gentle Persistence Towards Meaningful Change

When discussing how participants defined success in the Navigators’ practice, staff participants initially focussed on the key aim of the Navigator role to reduce alcohol-related hospital admissions. However, they acknowledged the complexity of the work carried out by the Recovery Navigators and the inherent challenge of capturing impacts beyond the standard quantitative performance measures. Persistence, advocacy, and proactiveness were viewed as fundamental to the Navigators’ practice as they seek to build relationships with patients and facilitate motivation development, as demonstrated by the following:

Sometimes it’s not always about bed days in hospital and reducing admission. Sometimes it is just about being there for the person, being that advocate. It’s hard to measure really. I think it is hard to measure the successes, because it’s all very qualitative really.(Sharon, Alcohol Care Team)

Participant accounts demonstrated how success for the Navigators can be subtle and immediate, or evolve over a long period of time. Both staff and patients further provided examples of transformational change for individuals in recovery, such as being reunited with children, transitioning from homelessness to stable housing, gaining financial control and security, developing new routines for meaningful abstinent living, and taking up work and caring responsibilities.

Patient participants’ accounts highlighted the key role played by the Navigators in terms of the choices the patients made (autonomy), the activities they engaged in (competence), and the necessity of the right kind of emotional and practical support (relatedness) in facilitating their decision making and working towards recovery goals. The interconnections between relatedness, autonomy, and competence are conveyed in the following quotes:

… when you’re in that position where you’re starting to feel worthless and alone, and you’re in there at your worst. And then when it actually feels as though people are looking out for you, it’s a good feeling. And obviously it’s just- It makes you think, “Right, I’ve got to sort things out now.”(Frank, patient, 50–55)

Yes, he’s got the support now and perhaps he was ready for it properly where he hasn’t been previously.(Anita, carer, 71–76)

In these accounts, both Frank and Anita appear to recognise a moment of “being ready” to make changes and the value of having access to support at that point. Frank conveys that the social connections were the catalyst for increased autonomy, while Anita appears to recognise that the individual in her care needed to be “properly” ready in themselves to fully benefit from support services. Together, these experiences demonstrate how autonomy, competence, and relatedness develop simultaneously; the individual and their social connections both matter and cannot be meaningfully separated.

The Navigators and their colleagues acknowledged that not all patients would be intrinsically motivated to change aspects of their lives or discuss their situation with healthcare professionals. While the Navigators talked about giving the patients space, their following accounts also illustrated that they had engaged in gentle persistence while also making patients aware that they were there when they needed them:

So I don’t just give up on somebody—If they’ve said they don’t want me I just pop my head round and say, “Hi, are you alright?” (Katie, Navigator)

They feel they’re in hell, basically, and they could be telling you to eff off, or worse. “Just go away. Just get out of my face—” Do you know what I mean? And then, it’s just sticking with them, and just like—I don’t know, like believing in them. And still giving them that hope, I suppose.(Jennifer, Navigator)

Katie and Jennifer appear to place value on engaging with patients and taking the time patients need to enable a therapeutic rapport to develop. Potentially, the key to the Navigators’ successful practice is about facilitating this unrushed process and laying a solid relational foundation.

### 3.4. Building Connections That Enable Change

While most of the participants agreed that the Navigators were better placed in the hospital setting than in community drug and alcohol services, all of the Navigators connected with a wide range of services across hospital and community settings to support patients in their longer-term recovery. These contacts and service links varied based on each Navigator’s cohort of patients’ specific needs. For example, one Navigator worked especially closely with probation officers, another with safeguarding, and another with homeless services.

Within this variation in local practice, participants described Navigators developing multiple working relationships across the health and social care system. Both patient and staff participants appeared to perceive the trusting relationships that the Navigators built with patients as essential for the further shared identification of support needs, followed by connecting with other practitioners and agencies in the best position to help. The patient and staff quotes below illustrate the Navigators’ contribution to continuity of care, flowing from relational to information and management continuity:

I think [the Navigator] has been sort of central to everything. But obviously she has come to see me when I’ve been at my worst. But then from that point she’s been able to signpost here or there. To have her at the centre of things, that’s been really good.(Barry, patient, 50–55)

I think that it’s created a new pathway for clients that we wouldn’t have been aware. For people that were already in services, it’s given them a new opportunity to have felt seen and heard and respected in medical services, which in turn is going to bolster their confidence and faith as well.(Jane, community provider)

The Navigators were valued by other staff in the NHS Trusts for their perceived ability to enable more appropriate care pathways for people who attend hospitals frequently due to alcohol use, and the interconnected challenges that have a bearing on their mental health and wellbeing. The following quote illustrates how a staff member perceived the Navigator was contributing to continuity of care through maintaining relationships with patients:

We have some of our regular attenders that will come in, and as soon as their name comes on the board… [Navigator] doesn’t wait for a referral. She doesn’t wait for a phone call. She just goes straight there, because she knows the person, because she’s already been dealing with them. And because she knows their background she’s able to get to the present-day problem quicker.(John, liaison service)

Some concerns were shared in addition to these positive accounts of the relationships that the Navigators had been able to develop by connecting patients with community organisations. In one NHS Trust, the participant accounts suggested that there was more work to be performed to facilitate shared working between the Navigators and community services. The participants suggested that early opportunities for effective data sharing across care settings were potentially missed, with potential ramifications to information continuity, as described by the following quote:

Obviously, there were data sharing agreements, put barriers in the way, and I don’t think they were thought of initially in terms of who and how and what information can be shared to ensure that those pathways are effective… the preparation prior to this role becoming live could have been discussed with partners, partner agencies in the community a lot sooner, so we could all have had a better understanding.(Catherine, community provider)

Another participant highlighted that the Navigators’ ability to signpost and refer to further support beyond the hospital was dependent on what resources were available in local services. The following quote illustrates the challenge of meeting patient needs where relevant community resources are scarce:

Well, there’s not enough housing, there’s not enough money. It’s like there’s not enough of anything. It’s all very well us doing a referral and saying that this patient’s homeless but there’s nowhere for them to go.(Lauren, Alcohol Care Team)

### 3.5. Creating the Conditions Where Navigators Can Flourish

Whilst we identified several transformational recovery stories in these interviews, it was also clear that the Navigators work in a challenging context. The Alcohol Care Teams (ACTs) and Navigator roles are relatively new initiatives for the region, and have been introduced in an existing landscape of severe alcohol harm across the NENC, as illustrated in this quote from a community-based service:

I think what we’ve seen is the amount of referrals we get now from the alcohol care team, some of those- Because these roles have just come, if you like, it’s too late for a lot of these people and we’ve seen a lot of deaths where people have been referred really poorly because that wasn’t in place 10 years ago, maybe, when somebody’s drinking was escalating to a hazardous level.(Lily, community provider)

Further challenges to the Navigators’ practice identified by staff participants related to concerns about the impact of uncertain funding provisions for both Alcohol Care Teams and Navigators. As the following quote suggests, this was viewed as short-sighted given the potential for cost-savings arising from effective alcohol harm prevention work:

You know, when it comes to how much alcohol-related illnesses and injuries actually contribute towards the NHS, I would imagine that having this buffer would be a huge reduction long-term…yeah, hopefully, they don’t get taken away.(Jane, community provider)

However, the staff members also reflected on the adverse impact that short-term funding could have on the Navigators themselves. Staff highlighted the great personal risk that the Navigators were taking to accept these fixed-term roles, which was perceived as having a tangible impact on service continuity for patients, as described in the following quote:

I think if we know that the role is going to be secure that is much better. Because I know quite a lot of Recovery Navigators in hospitals have already moved on, because they don’t know their job is secure.(Sharon, Alcohol Care Team)

Alongside these structural factors, staff participants also described the challenging nature of the work itself. Navigators’ and their ACT colleagues’ routine practice involved dealing with antagonistic situations with intoxicated patients, multifaceted safeguarding issues, and the grief arising from having developed relationships with patients who eventually died due to alcohol-related harm. This type of work can elicit strong emotional responses. Zoes’s comment illustrates these everyday realities of working with the patient group:

And we do sometimes have to have difficult conversations. We need to understand and listen to complex traumas that people have been through. That can be quite uncomfortable and upsetting.(Zoe, specialist nurse)

We found that the Alcohol Care Teams were offering emotional and practical support to the Navigators, providing a vital source of support that extended beyond formal supervision and debrief. As Jennifer’s comment below suggests, the Navigators we interviewed highlighted the value of this support in helping them feel ‘part of the team’:

Lots of my patients have died since I’ve started. So, to bounce each other, and laugh, and cry.(Jennifer, Recovery Navigator)

The accounts suggested that this support was evident from the outset: when the Navigator roles were introduced, the Alcohol Care Team members played a key role in preparing them to work in the unfamiliar hospital environment, as described in the following quote:

… for me it was about how I ensure the well-being of putting a non-clinical person into [the ED] where it’s all hands to the pumps. It was basic things like what uniform do they wear? Can they be in civvy clothes.(Violet, Alcohol Care Team)

## 4. Discussion

Navigator roles have been introduced in the North East and North Cumbria (NENC) Alcohol Care Teams (ACTs) to better support the post-discharge recovery of people who attend hospitals frequently for alcohol-related reasons. We applied the concept of continuity of care (CoC, [36,37]) and Self-Determination Theory (SDT, [38]) to help make sense of our qualitative data and to connect our findings to the wider theoretical and conceptual landscape. Our aim was to understand the Navigators’ contributions to alcohol care and recovery by interviewing the Navigators, their cross-sectoral colleagues, patients, and carers. Essentially, the experiences that the participants shared were stories of human relationships within care pathways and service configurations: building trust, listening and being listened to, persisting in the face of challenges, and not giving up on the possibility of a better future. Patients’ recovery trajectories were about developing individual agency and motivation through meaningful social connection with the Navigators and their colleagues. The Navigators built relationships proactively with patients, carers, and colleagues. While there was local variation, in each area, the Navigators had had successful experiences in developing networks that helped them to practice safely and for patients to have an active plan on who to contact and where to go when leaving the hospital.

Across geographical and service contexts, Navigator services are founded on person-centred care and respectful, collaborative relationships with patients [44,45]. Navigator roles have the potential to reduce barriers for people to obtain support with their health and social needs [46]. Other studies have found that locating and accessing the right support in a complicated care system is a commonly shared experience among people who are supported by the Navigators [47,48,49]. Our qualitative study builds on this work and contributes further evidence from the context of alcohol care in acute hospitals in the NENC. Within the acute hospital environment, the participants shared that the Navigators have helped to adapt the ED and hospital to places where patients can better find social connection. Drawing on our conceptual framework of SDT [39] and the concept of CoC [37,38], we have highlighted that this relatedness was perceived as contributing to both individual patients’ motivation development and relational continuity. Most of the Navigators described being supported by their managers and colleagues to work with a small group of patients so that they could work with the necessary depth to build strong relational foundations.

A key, albeit unsurprising, finding was the importance of robust management, supervision, and collegial support to maintain the Navigators’ safe practice and their own wellbeing. Across the UK, link workers have reported concerns about their own competence to work autonomously with people who require more specialised support [50]. In the NENC, Navigator roles were designed to be embedded and managed within the ACTs. It was evident that the participants viewed the Navigators as the part of the ACT, and the ACT as the Navigators’ team; both were mentioned together in many of the participants’ responses. The Navigators appeared to add meaningfully to the ACTs, rather than operate as a fully autonomous service. Although our participants identified gaps in wider services, such as accommodation for homeless patients, it appears that the Navigators are well-placed within the ACTs to work within their competence and to seek emotional and professional support from their colleagues as required.

Patients and colleagues valued having a Navigator working directly in the hospital, even though the NENC Navigator roles were originally designed to combine assertive outreach in the community alongside hospital-based work. However, assertive alcohol outreach that provides intense, frequent, and longer-term support [51] requires a team, not a single practitioner. Similarly with peer workers in alcohol and other substance use rehabilitation [31], the Navigators appear to “add a layer” to the existing services and offer a connection point between hospital- and community-based services. If Navigator roles are transitioned from being based in hospitals towards working across a wider geography, preserving this additional value might be a key consideration.

Working in the interface of hospital- and community-based care, primarily within the hospital and its organisational culture, is not without its challenges. Our findings from ACTs and community provider interviews point towards tensions between the narrow focus of hospital performance measures (e.g., the length of stay) and the complex, nuanced, and iterative nature of recovery processes and outcomes. Interestingly, the Navigators we interviewed did not bring up performance-related pressures, but primarily talked about their relationships with patients, carers, and multidisciplinary colleagues. Potentially, in the hospital context, the clinically senior ACT colleagues are more acutely aware of performance-driven pressures than the Navigators are. Psychiatric liaison nurses, with similar educational and clinical seniority levels with ACT nurses and managers, have critiqued the standards that provide a performance framework for their practice [52]. However, our findings also contrast with some of the link worker literature that identifies a tension between a system-level demand for a high rate of onward referrals as a key signposting outcome and link workers’ hopes to offer more comprehensive support to the people they encounter in their practice [50,53,54]. These practitioners work within similar roles and qualifications as the Navigators in our study.

Recovery is a long-term process that incorporates changes in the individual’s identity and daily activities, and their roles and relationships with family members and significant others, social groups, and the wider society [55,56]. From the point of view of services, it is a multi-dimensional construct that services operationalise in different ways [57,58,59,60]. Through the lens of SDT, outcomes can be seen as reflecting patients’ motivational states, choices, circumstances, and experiences. Thus, social connection with the Navigators and other professionals is a part of the key context for motivation development through fulfilled basic psychological needs. Ideally, performance measurements can reflect this understanding of patients as people who develop relationships and build their futures as they travel through services across the care system. While most services use patient-reported outcome and experience measures to evaluate an aspect of a service at one time point, there are international examples of these measures being used for continuous treatment planning and service improvement as a part of routine care [61]. Levesque and Sutherland [62] propose that, within an integrated performance measurement framework, healthcare performance areas should be measured in relation to each other. This appears as a relevant consideration for the Navigators and their link worker colleagues across services, due to their roles as building connections across various services, to facilitate outcomes that matter to people who access this support.

While the existence of ACTs and the Navigator roles within them appear to represent a promising development for many patients, carers, and addiction professionals, it is worth remembering that ACTs were a policy response to try to address alcohol-related harm in the most severely impacted regions of England. The work is emotionally challenging, and not all of the patients live long enough to develop the internal motivation towards health-enabling change. In wider link worker and peer worker literature, there is a common thread of identifying the importance of self-care and developing boundaries that support the practitioner’s own wellbeing, and to facilitate coping with emotionally challenging work [30,31,63,64].

Furthermore, in addition to the relational, information, and management dimensions, robust CoC requires a context of service continuity where the Navigators, ACTs, and their colleagues can plan and carry out their work within longer timeframes than 12-month funding periods. The strategic ambition to improve the health and wellbeing of people who attend hospitals frequently for alcohol-related reasons should focus on the continuous improvement of the care that ACTs and the Navigators within them provide. A longer-term funding solution would enable more robust planning and setting high expectations for the contributions that Navigators and ACTs make. Importantly, the secured existence of these teams and roles might contribute positively to patients’ and carers’ psychological wellbeing, as short-term funding indicates that support might come to an abrupt end.

The key limitations of our study concern sampling and recruitment. We intended to carry out our study across six hospitals and their surrounding services and areas in the NENC. However, two of the hospitals had Navigator vacancies for a part of or throughout the study, which resulted in fewer interviews across the full study geography than we had intended. Despite the proactive efforts to enhance the recruitment across all of the remaining sites, including the involvement and suggestions from our PPIE groups, we recruited fewer patients and carers than originally intended. In reflexive thematic analysis, the meaning of the data is generated rather than discovered, and sample size and data saturation develop through ongoing decision making [65]. We aimed at the maximum variation in participants’ backgrounds and experiences, and interviewed a wide range of professionals with diverse career paths, providing that they met the inclusion criteria of being Navigators or working closely with them. Had we managed to listen to more carers within the study’s timeframe, our analysis might have gained new depth and offered us an avenue to gain insights about, e.g., the Navigators’ roles in carer health and wellbeing, and negotiating Navigator roles in the dynamics between patients, carers, and wider services. Importantly, the patients who engaged with our research may have been the patients who had benefited from Navigator support the most. The absence of more dissenting voices in our interviews does not mean that they do not exist.

Future studies evaluating similar services and populations might achieve more variation in participants’ experiences and increasingly nuanced data through different choices on research methods and approaches. For example, [66] used ethnography to study hospital staff alcohol care cultures, while service users in acute mental health inpatient wards have shared their experiences of an occupational therapy department through photo elicitation [67], and new insights about the complexity of medication reviews have been developed through using video recordings from practice in reflexive workshops [68]. A study design that would have permitted the researchers to be embedded in the Navigators’, ACTs’, patients’, and colleagues’ real-world contexts, with careful ethical and practical decision making around the methods, might have allowed us to enrich our data collection. Potentially, future research in similar contexts can apply this learning to reach different groups of patients, including those with more pressing mental health concerns, highly distressing life circumstances, low or no motivation towards changing drinking patterns, or less positive care experiences with practitioners and services.

Our study’s focus was on developing qualitative insights about a regionally implemented role. However, we can offer some recommendations for practice and policy in similar contexts or with similar patient populations. Regardless of the differences in the job specifications and service locations, all of the Navigator roles in our study had protected time to work with a small caseload. This enabled them to provide a person-centred service, with the potential to build trusting relationships with patients over time. The Navigators’ commitment to individually tailored care and the ability to connect with various other practitioners was supported by the commitment of their managers to value listening and thoroughness as key indicators of the Navigators’ success. We encourage managers and commissioners to consider the trade-offs between the volume of patients seen and the comprehensiveness of care provided when similar roles are implemented in any context that involves supporting people who access a service frequently without apparent improvements in their situation over time. Furthermore, Navigator roles do not need to be psychologically specialised to be psychologically demanding. Our study gives an example of highly adaptable Navigators that are able to look after their own wellbeing and maintain safe practice through robust communications with their team members. Again, it is important to consider what kind of environment a new practitioner is entering when these kinds of roles are set up. Encouraged by the example of the Navigators in the NENC ACTs, we recommend that Navigators and link workers join a team where they work autonomously and are supported, instead of becoming autonomous individuals within organisations and care systems. This is especially important considering the care of people who attend hospitals frequently for alcohol-related reasons and people in similar cohorts who might go through many doors before encountering someone who listens and makes their best effort to help.

## 5. Conclusions

Navigators can improve CoC for people who attend hospitals frequently for alcohol-related reasons through working directly in hospitals and proactively at the interface of a range of community- and hospital-based services. While our study sought to be specific to its NENC context, the insights we have generated can inform the development and evaluation of similar roles in national and international alcohol care, as a part of health service efforts to address and reduce alcohol-related health inequalities. Navigators also contribute to the wider workforce of link and peer workers, sharing many of the challenges and strengths with these cross-sectoral colleagues. People in these roles build connections across services to enable meaningful recovery processes, rooted in meeting patients’ and service users’ needs for relatedness, competence, and autonomy. Strong supervision and management support is key to the successful delivery of these roles. To deliver person-centred support and facilitate continuity of care, people in these roles need dedicated time to work with a small number of people with multiple health and social support needs.

## Data Availability

The data that support the findings of this study are available on request from the corresponding author. The data are not publicly available due to privacy and ethical restrictions.

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
