# Peer review of "Building Connections and Striving to Build Better Futures: A Qualitative Interview Study of Alcohol Recovery Navigators’ Practice in the North East of England, UK"

_ijerph, 2025, doi:10.3390/ijerph22010111_

Round 1
Reviewer 1 Report
Comments and Suggestions for Authors
Dear Authors
I congratulate you on taking on a difficult and important topic for the health of the population.
However, I would ask you to consider introducing a few additional threads:
1. The research sample is very small, the study needs to be repeated on a larger sample;
2. The article does not contain any tables or summaries, and in the material and methods area it is not clearly stated what research questions it answers and what the distribution of answers is;
3. In the results area, instead of loose statements, tables with the questions asked should be provided and the distribution of answers should be analyzed;
4. Apart from a thorough review of opinions and a review of literature related to the topic, there is no clearly defined purpose of the study;
5. The assessment of clinical and economic effectiveness should be examined based on a structured questionnaire and on a statistically significant sample; the obtained results should be subjected to a statistical analysis confirming the correctness and statistical significance of the data obtained.
Dear Authors
Your article is a collection of clinical cases, if it is to serve to assess the effects of introducing Navigators, it should be constructed as a retrospective analysis. In its current form, it is closest to being a short report or a literature review in the form of a [pilot evaluation.
Reviewer 2 Report
Comments and Suggestions for Authors
This is a very interesting study and provides insights into the Navigator role in NENC. The study would benefit from an understanding of the training/CPD that is required to sustain these Navigator roles. Hence, some discussion on recruitment of Navigator roles from wider international studies could be added.
In terms of the theoretical underpinning, a brief outline could be added in the methods section re SDT and a summary, in table form perhaps, in the results/discussion with brief narratives from the stakeholders. Overall this is a concise account of the key stakeholder experiences on building connections in Navigator roles on Alcohol Recovery.
Reviewer 3 Report
Comments and Suggestions for Authors
This paper is a qualitative study addressing the contributions Alcohol Recovery Navigators make to integrated alcohol care. The paper is very well-written and conducted thoroughly. I have a few suggestions to improve the paper:
Abstract: Please add a brief explanation of the role of Recovery Navigators.
Introduction:
Line 43: When describing the numbers who drink at high levels, percentages would be more helpful.
The introduction is very well written and flows logically. However, I would like to have seen some discussion about health inequalities and alcohol recovery.
Methods:
Theoretical approaches
It is strange that theory is mentioned in the Method section. I would move it to the Introduction, to set the context.
In general, the Methods section is very thorough and detailed. Excellent discussion of reflexivity and PPIE involvement.
Results
Lines 249-50: Please edit to read: “The age range of the patient and carer group was 34-69 years …”
The Results section is very well written – it provides deep insight into the issues involved.
Discussion
The first paragraph should include a brief summary of the findings, before you discuss the issues in more depth.
Line 553-54 – Reword to read: “Patients and colleagues valued having a Navigator working directly in the hospital even though the NENC …”
Line 576-577 – Reword to read: “… link workers’ hopes to offer more comprehensive support to people they encounter in their practice…”
Line 607 – Edit “cope” to “facilitate coping”
Overall, the Discussion provides excellent discussion of the findings.
I would like to see another paragraph that addresses the implications for policy and practice.
Round 2
Reviewer 3 Report
Comments and Suggestions for Authors
I would like to thank the authors for the changes they have made to the manuscript. These changes have greatly improved the manuscript and I have no further comments.